# Effects of Drought Stress and Rehydration on Physiological and Biochemical Properties of Four Oak Species in China

**DOI:** 10.3390/plants11050679

**Published:** 2022-03-02

**Authors:** Shifa Xiong, Yangdong Wang, Yicun Chen, Ming Gao, Yunxiao Zhao, Liwen Wu

**Affiliations:** 1State Key Laboratory of Tree Genetics and Breeding, Chinese Academy of Forestry, Beijing 100091, China; xiongshifa@163.com (S.X.); wyd11111@126.com (Y.W.); chenyc@caf.ac.cn (Y.C.); 4862705@163.com (M.G.); zyx_yunxiao@163.com (Y.Z.); 2Research Institute of Subtropical Forestry, Chinese Academy of Forestry, Hangzhou 311400, China

**Keywords:** drought stress, water parameters osmotic solutes, antioxidant enzyme, photosynthesiss, *Quercus*

## Abstract

*Quercus fabri* Hance, *Quercus serrata* Thunb, *Quercus acutissima* Carruth, and *Quercus variabilis* BL are four Chinese oak species commonly used for forestation. To ensure the survival of seedlings, we first need to understand the differences in drought resistance of the four oak species at the seedling stage, and comprehensively evaluate their drought resistance capabilities. The four oak seedlings were divided into drought-rewatering treatment group and well watered samples (control group). For the seedlings of the drought-rewatering treatment group, drought stress lasting 31 days was used, and then re-watering for 5 days. The water parameters, osmotic solutes content, antioxidant enzyme activity and photosynthesis parameters of the seedlings in the two groups were measured every 5 days. Compared with the control group, the relative water content, water potential, net photosynthetic rate, transpiration rate, and stomatal conductance levels of the four oaks all showed a downward trend under continuous drought stress, and showed an upward trend after rehydration. The soluble protein, soluble sugar, proline, peroxidase, superoxide dismutase and catalase content of the four oaks increased first and then decreased under drought stress, and then increased after rehydration. The content of glycine betaine and malondialdehyde continued to increase, and gradually decreased after rehydration. The weight of each index was calculated by principal component analysis, and then the comprehensive evaluation of each index was carried out through the membership function method. The drought resistance levels of the four oak species were as follows: *Q. serrata* > *Q. fabri* > *Q. variabilis* > *Q. acutissima*.

## 1. Introduction

With rising global temperatures and frequent extreme phenomena, 64% of the world’s land is already under drought stress, which has seriously affected the normal growth and development of many plants and crops [1]. Half of China’s land is arid or semiarid, and drought is a serious problem in northwest and southwest China [2]. Moreover, even southern regions with abundant rainfall are often affected by seasonal drought, which is a major factor that limits the normal growth and development of plants [3].

When plants are under drought stress, the plant water balance is severely affected, and normal physiological processes are compromised [4]. The relative water content (RWC) and water potential of plant leaves decrease [5]. In addition, plants close their stomata to reduce water loss, which can limit the supply of CO_2_ and lead to reduced photosynthesis Reactive oxygen species (ROS) metabolism is the primary response of plants to stress [6]. Drought stress can lead to the disorder of production and elimination of ROS in plants, and excessive accumulation of ROS causes oxidative stress [7]. To protect themselves from oxidative stress, plants produce antioxidant enzymes and nonenzymatic substances that scavenge ROS [8]. The antioxidant enzymes mainly include peroxidase (POD), superoxide dismutase (SOD), ascorbate oxidase (APX) and catalase (CAT), and the nonenzymatic antioxidant substances mainly include reduced glutathione (GSH), reduced ascorbic acid (AsA) and carotenoids [9]. Many studies have shown that the responses of different protective enzymes to drought stress vary [10,11,12]. The protective enzymes that play a dominant role in plants may also differ between periods of stress [13]. In addition, plants can increase the concentration of cell fluid, reduce the osmotic potential, maintain high water absorption and maintain the stability of turgor pressure by accumulating osmotic regulators [14].

The morphological structure (leaf area, leaf thickness and stomatal density) and physiological properties (RWC, antioxidant system and osmotic solutes) of leaves are often used as important indicators in drought studies [15,16,17]. At present, many studies have been carried out on the physiological and biochemical responses and drought resistance mechanisms of plants under drought stress [18,19,20]. A pot experiment with artificially controlled water was performed to simulate the natural drought process in the soil and to observe the plant resistance response to different levels of drought stress [21,22]. Pot experiments more objectively reflect the response characteristics of plants to natural drought stress. There are many methods to evaluate the drought resistance of plants, such as the affiliation function method, hierarchical evaluation, grey correlation analysis and principal component analysis [23,24,25]. Among them, the affiliation function method can make a more comprehensive and integrated evaluation based on the determination of multiple indicators, avoiding the limitations and inaccuracies of a single evaluation indicator, and is the most widely used in drought resistance evaluation [26,27].Recently, there has been a preference for combining multiple evaluation methods to improve the accuracy of the results [28]. Seedlings are sensitive to drought stress at the beginning of their development, which not only threatens their survival but also affects their later growth, biomass formation and ability to overwinter [29]. Therefore, by studying the physiological and biochemical characteristics of oak seedling leaves under drought stress and combining various evaluation methods, we can evaluate drought resistance with greater reliability.

The area of *Quercus* accounts for 13.7% of the total area of natural forests in China and is the main component of natural forests in China [30]. The *Quercus* plant genus plays an important role in water conservation, soil and water conservation and ecological stability [31]. *Quercus fabri* Hance, *Quercus serrata* Thunb, *Quercus acutissima* Carruth and *Quercus variabilis* Bl are the dominant species in deciduous broad-leaved forests and mixed coniferous forests in temperate and warm temperate regions of China and are also the forest building blocks in most parts of the country [32,33]. Oak tree wood is dense and beautifully textured, resistant to wear and decay and is often used as a high-quality material for building and household purposes [34]. Oak tree bark can be used for extracting tannins, young leaves can be used to raise sericulture, and dead branches can also be used to cultivate valuable medicinal herbs such as edible mushrooms and ganoderma lucidum [35,36]. Oak has high economic value and is rich in oak resources in China, but its development and utilization is still backward. At present, most research on oak trees has focused on the nutritional and mineral contents of the fruit, the distribution patterns of populations in individual areas and genetic diversity [37,38,39]. However, in the process of artificial afforestation, seedlings transplanted to mountains often suffer from seasonal drought, which seriously affects growth and development. This leads to a low survival rate and poor afforestation effects. Understanding how drought resistance varies across different oak species is helpful for reasonable cultivation and extension. In this study, four common oak species were selected, and the photosynthetic capacity, osmotic solutes content and antioxidant enzyme activity of the leaves were studied under short-term natural drought stress and rehydration processes. The differences in the response mechanisms of the four oak species to drought stress were preliminarily investigated, and drought resistance was comprehensively evaluated to provide a theoretical basis for extensive cultivation and promotion.

## 2. Results

### 2.1. Soil Moisture Content

The soil moisture content of the four oak control groups ranged from 28.0% to 31.9%. With continuous drought stress, the soil moisture content of the treatment group decreased gradually, and the range of the decrease was similar. The soil moisture content reached the lowest point of 3.1~3.5% at day 31 of drought stress and returned to the control level after rewatering (Figure 1).

### 2.2. Water Potential and RWC

The RWC and water potential of the leaves of all four oak species tended to decrease under continuous drought stress compared to those of the control group. The values of the treated groups were not significantly different from those of the control group on the first day, but all decreased and reached the lowest values, which were significantly lower than those of the control group, on day 31 (*p* < 0.05). After rewatering, the leaf water potential increased rapidly, and the recovery rate of the leaf water potential was faster than that of RWC on day 36. The RWC and water potential of *Q. fabri* leaves decreased from 81.13% and −1.52 MPa (day 1) to 41.10% and −2.84 MPa (day 31), with decrease rates of 49.47% and 87.25%, respectively. After rewatering, the RWC and water potential recovered to 73.02% and −1.67 MPa (day 36), respectively, but did not recover to the control sample level. The RWC and water potential of *Q. serrata* leaves decreased from 81.67% and −1.48 MPa (day 1) to 48.67% and −2.45 MPa (day 31), with decrease rates of 40.41% and 66.14%, respectively. After rewatering, the RWC and water potential recovered to 76.02% and −1.52 MPa (day 36), respectively. The RWC and water potential of *Q. acutissima* leaves decreased from 81.72% and −1.43 MPa (day 1) to 43.23% and −2.46 MPa (day 31), with decrease rates of 47.10% and 72.03%, respectively. After rewatering, the RWC and water potential recovered to 72.72% and −1.54 MPa (day 36), respectively. The RWC and water potential of *Q. variabilis* leaves decreased from 81.02% and −1.52 MPa (day 1) to 38.07% and −2.48 MPa (day 31), with a decrease rate of 53.10% and 63.16%, respectively. After rewatering, the RWC and water potential recovered to 73.67% and −1.57 MPa (day 36), respectively. The leaf water potential of *Q. serrata*, *Q. acutissima* and *Q. variabilis* returned to control samples level, but the RWC remained lower than the control sample level (Figure 2).

### 2.3. Osmotic Solutes

Under continuous drought stress, the levels of soluble protein, soluble sugar and proline of the four oak species first increased, then decreased, and then increased after rehydration. However, the levels of glycine betaine increased continuously and decreased after rehydration. The SP levels of *Q. fabri*, *Q. acutissima* and *Q. variabilis* reached a peak (day 6) earlier than that of *Q. serrata* (day 11), which was 1.50, 1.28, 1.15 and 1.11 times higher than that of the control group, respectively. Then, the SP levels of the four oak species were the lowest on day 31 and were significantly lower than that of the control group (*p* < 0.05). The SP levels were decreased by 43.74%, 48.75%, 42.52% and 51.60%, respectively. After rewatering, only *Q. fabri* returned to control sample level, while the other three oak species were significantly lower than control samples (*p* < 0.05) (Figure 3A). The SS levels of the four oak species reached a peak (day 26), which was 1.46, 1.48, 1.35 and 1.37 times higher than that of the control group, respectively. Then, the SS levels decreased and were the lowest on day 31, decreasing by 15.44%, 12.79%, 10.62% and 14.22%, respectively. After rewatering, the SS level of *Q. serrata* were significantly lower than the control samples (*p* < 0.05), but the other three oak species returned to control samples level (Figure 3B). The Pro levels of the four oak species reached a peak (day 26), which was 1.89, 1.66, 1.98 and 1.87 times higher than that of the control group, respectively. Then, the Pro levels decreased but was still significantly increased compared with the level in the control group (day 31) (*p* < 0.05) and returned to the control sample level after rehydration (Figure 3C). The GB levels of the four oak species showed a continuous increasing trend, reaching the peak on day 31, which was was 1.56, 2.05, 1.96 and 2.29 times higher than that of the control group, respectively. After rehydration, the GB levels decreased sharply but was still significantly higher than that in the control group (*p* < 0.05) (Figure 3D).

### 2.4. Antioxidant Enzyme Activities and MDA

Under continuous drought stress, the activities of POD, SOD and CAT of the four oak species first increased, then decreased, and then increased after rehydration. The MDA level increased continuously and decreased after rehydration. The POD activity of *Q. fabri*, *Q. acutissima* and *Q. variabilis* (day 16) reached a peak later than that of *Q. serrata* (day 11), which was 1.36, 1.11, 1.34 and 1.18 times higher than that of the control group, respectively. The POD activity levels of the four oak species were the lowest on day 31 and were significantly lower than that of the control group (*p* < 0.05). The POD activity levels were decreased by 18.34%, 32.21%, 34.74% and 36.65%, respectively, and did not return to control samples level after rehydration (Figure 4A). The SOD activity level of the four oak species was significantly different from that of the control group at day 5 (*p* < 0.05) and peaked at day 21. The SOD activity levels were 2.14, 2.33, 1.93 and 2.28 times higher than that of the control group, respectively. Then, the SOD activity levels were the lowest at day 31 and were significantly lower than that in the control group (*p* < 0.05). The SOD activity levels were reduced by 32.55%, 29.26%, 37.10% and 20.73%, respectively, and did not recover to control samples level after rehydration (Figure 4B). The CAT activity levels of *Q. fabri*, *Q. acutissima* and *Q. variabilis* (day 21) reached a peak later than that of *Q. serrata* (day 16). The CAT activity levels were 1.54, 1.48, 1.52 and 1.39 times higher than that of the control group, respectively. The CAT activity levels of the four oak species were the lowest on day 31 and were significantly lower than that of the control group (*p* < 0.05). The CAT activity levels decreased by 25.13%, 20.61%, 8.12% and 11.43%, respectively. After rehydration, only *Q. fabri* returned to the CAT activity level of the control sample, but the levels of the other three oaks were significantly lower than control samples (*p* < 0.05) (Figure 4C). The MDA levels of the four oak species showed a continuous growth trend, and there were no significant differences in the levels at the early stage (day 1 and day 6) when comparing between the four oak species and the control group (*p* > 0.05). The MDA levels peaked on day 31 and were significantly higher than that in the control group (*p* < 0.05), increasing by 2.71, 2.15, 2.23 and 2.06 times, respectively. The MDA levels decreased sharply after rehydration but were still significantly higher than that in the control group (*p* < 0.05), increasing by 1.38, 1.41, 1.36 and 1.45 times, respectively (Figure 4D).

### 2.5. Photosynthetic Parameters

Compared with the control group, the Pn, Gs and Tr values for the four oak species showed a decreasing trend under continuous drought stress. The values were the lowest on day 31 and rose rapidly after rehydration. As drought stress was prolonged, the Pn values of the four oak species decreased continuously. The Pn values of *Q. serrata* at day 6 was significantly higher than that of the control group (*p* < 0.05). The Pn values were decreased by 72.10%, 63.29%, 63.67% and 60.16%, respectively (day 31). After rehydrating, the Pn of *Q. serrata* was significantly lower than control samples value (*p* < 0.05), but the Pn values of the other three oak species returned to control samples (Figure 5A). There were no significant differences in the Gs values when comparing between the four oak species and the control group at the early stage (day 1 and day 6) (*p* > 0.05). The Gs values were decreased by 71.14%, 57.45%, 50.03% and 45.24%, respectively (day 31), but returned to control samples values after rehydration (Figure 5B). There were significant differences in the Tr values when comparing between the four oak species and the control group at day 6 (*p* < 0.05). The Tr values were decreased by 62.37%, 60.08%, 61.28% and 60.72%, respectively (day 31), and did not return to control samples values after rehydration (Figure 5C).

### 2.6. Multivariate Statistical Analysis

The eigenvalues and contribution rates of principal components were the basis for selecting principal components. The 13 physiological and biochemical indexes of the leaves from the 4 oak species were analysed by PCA. Two principal components with eigenvalues greater than 1 were obtained, and their contribution rates were 64.24% and 19.70%, respectively. The cumulative contribution rate was 83.94%, and most of the information on the original characteristics was retained (Table 1). Therefore, the first two principal components could be selected as the important principal components of the drought resistance of the four oak species. The factors with higher loading capacity in the first principal component were Ψw, RWC, Pro, SP, GB, MDA, Pn, GS and Tr, which were mainly related to leaf water status, osmotic solutes and photosynthesis. The second principal component was mainly related to antioxidant enzymes. The treatments of the 4 oaks under drought stress and during rehydration were completely separated. In addition, *Q. acutissima*, *Q. serrata* and *Q. variabilis* highly overlapped under drought stress and during rehydration. In contrast, *Q. fabri* varied greatly with the other three species (Figure 6). To further understand the relationship between leaf water status, osmotic solutes, photosynthesis and antioxidant enzymes, Pearson correlation analysis was used to analyse the data. The results showed that Pn, GS and Tr positively correlated with each other, negatively correlated with MDA, GB and Pro levels and positively correlated with SP, Ψw and RWC. Ψw and RWC levels were significantly negatively correlated with Pro, GB and MDA levels. MDA levels were significantly positively correlated with GB and Pro levels and negatively correlated with SP levels and POD activities. Pro levels were significantly negatively correlated with SP levels and positively correlated with SS and GB levels. In addition, SOD, POD and CAT activities were significantly positively correlated (Figure 7). To comprehensively evaluate the drought resistance of the four oak species, 13 physiological and biochemical indexes of the four oak species under continuous drought stress and rehydration conditions were analysed by membership functions (Table 2). The membership function value of each index was calculated according to a formula. The degree of correlation between different indexes and drought resistance was different. The arithmetic mean of the membership value of each index did not fully reflect the level of drought resistance in the four oak species. Therefore, the proportion of eigenvalues corresponding to each principal component to the sum of the total eigenvalues of the extracted principal components was taken as the weight. The comprehensive evaluation value of the four oak species was calculated. Higher comprehensive evaluation values were positively correlated with stronger drought-resistance ability. The results showed that the order of drought resistance of the four oak species were as follows: *Q. serrata* > *Q. fabri* > *Q. variabilis* > *Q. acutissima*.

## 3. Discussion

### 3.1. Leaf Moisture Status

Drought stress can lead to a decrease in the RWC and water potential of plant leaves [40]. It is generally believed that smaller decreases in the RWC and water potential of plant leaves correlate with higher water-retaining ability and stronger adaptability of the leaves to drought stress [41]. In our research, under continuous drought stress, the relative water content and water potential of the four oak species decreased to different degrees, which indicated that the four oak species could absorb more water by reducing Ψw to resist drought stress. Among them, the RWC of *Q. serrata* declined the least and recovered the fastest after rehydration. *Q. variabilis* showed the fastest decrease in RWC but the smallest decrease in Ψw. The Ψw of *Q. fabri* decreased the most and recovered the slowest after rehydration. The decline of water status of four oak species in different degrees is related to their unique stomatal control and cuticular tranpiration. These results indicate that *Q. serrata* can maintain a higher water balance than the other three oak species. *Q. serrata* leaves may be more leathery than the leaves from the other species, which would be beneficial for reducing water transpiration during drought stress. The results herein were similar to those from studies of soybean [42] and *Cyamopsis tetragonoloba* (L.) Taub. [43].

### 3.2. Osmotic Solutes

Plants under drought stress are often damaged by osmotic stress. When cells lose water, it can decrease turgor pressure and lead to death [44]. However, plants can actively accumulate osmotic solutes to maintain turgor [45]. In our research, the SS, Pro, GB and SP levels of the four oak species all showed an increasing trend in the early stages of sustained drought stress. The results showed that the four oak species reduced their osmotic potential by accumulating osmotic solutesduring early stages of drought stress to ensure that water could be absorbed from the external environment and to resist drought-induced damage. Soluble proteins respond faster than other osmotic regulators, it was significantly higher than that in the control group at day 6, similar to the results of Cheng et al. [46]. However, osmotic regulation cannot unlimitedly regulate the osmotic pressure of cells. When drought stress reached the limit of plant self-regulation, we found that the levels of Pro and SP of the four oak species began to decline at day 26. In particular, SP was more sensitive to drought stress and began to decrease earlier. It is possible that the stress damage exceeded the tolerance limit, which blocked normal plant metabolism and then affected the production of osmotic solutes. Similar results were observed in *Capsicum* spp. [47], in which the contents of SP and SS decreased under severe drought stress. At present, a large number of studies have shown that GB has multiple functions in resisting drought stress, including scavenging ROS, maintaining the stability of biofilms, and protecting the photosynthetic apparatus [48,49]. In this study, we found that GB increased continuously in the four oak species, indicating that they could stimulate their own GB synthesis under drought stress to resist the damage caused by stress. After rehydration, the levels of SP, SS and Pro returned to control samples in *Q. fabri*, and the levels of SS and Pro returned to control samples in *Q. acutissima* and *Q. variabilis*. However, only the Pro level returned to the control samples in *Q. serrata*. This also reflects the difference in the recovery ability of osmotic regulation system among the four oak species.

### 3.3. Antioxidant Enzyme Activities and MDA

Under normal conditions, there is a dynamic equilibrium between the production and clearance of ROS in plants, but when plants are under drought stress, the dynamic equilibrium is disrupted [50]. The excessive accumulation of ROS can damage cells and cause oxidative deterioration of cell membranes, which may lead to plant death in severe cases. There is an active oxygen scavenging system in plants, in which SOD, POD and CAT are important antioxidant enzymes for scavenging ROS [51]. In our research, the activities of POD, SOD and CAT enzymes in the four oak species showed a trend of increasing first and then decreasing. The results showed that all four oak species could resist the damage caused by drought stress by enhancing the activity of antioxidant enzymes. However, as the drought stress was intensified, the ROS production levels exceeded the scavenging abilities of the plants, and the activities of the three protective enzymes all decreased to varying degrees. Similar results were observed in *Handeliodendron bodinieri* (Levl.) [52], *Olea europaea* L. [53]. The activity peaks of SOD, POD and CAT appeared at different times in four oak species, which indicated that they had different sensitivity to drought stress. We also found that the SOD activity increased significantly at day 6. The results showed that SOD was the first to respond to the early stages of drought stress, and then POD and CAT enzymes responded. Compared with the control group, there were no significant differences in the MDA levels of the four oak species at day 6. The results indicated that the three enzymes could effectively eliminate the production of reactive oxygen species in the early stages of drought stress. However, in the late stages of drought stress, the MDA levels in the four oak species were significantly higher than that of the control group (*p* < 0.05). It is possible that the four oak species suffered from long-term drought stress, which damaged the antioxidant enzyme system to different degrees. After rehydration, the MDA levels still failed to fall to the control sample level, which indicates that the continuous drought stress has caused irreversible damage. ROS overaccumulation have destroyed the cell membranes, which may have caused MDA to accumulate in large quantities.

### 3.4. Photosynthetic Parameters

Drought stress can also affect the photosynthetic physiology of plant leaves, which is mainly reflected by the chlorophyll content, Pn and Tr [54]. In this study, the Pn, Gs and Tr values of the four oak species showed a continuous decreasing trend as drought stress was prolonged and were significantly lower than those of the control group at day 31 (*p* < 0.05). This result shows that photosynthesis was inhibited in the four oak species under continuous drought stress, which might be due to the production and accumulation of a large number of reactive oxygen species in the leaves. Accumulation of ROS results in damage to the mesophyll cell membrane, destruction of photosynthetic enzyme activity and a decrease in photosynthesis. Correlation analysis results also showed that photosynthetic parameters were significantly positively correlated with water status and negatively correlated with MDA content. In addition, the Pn, Gs and Tr values of *Q. fabri* were the lowest among the four oak species, the Pn and Tr values of *Q. serrata* were the highest, and the Gs value of *Q. variabilis* was the highest at day 31. The results showed that the resistance of *Q. fabri* photosystems was the worst, and *Q. serrata* was the strongest. It may be that the osmoregulation (especially the GB content) of *Q. serrata* is stronger than that of the other three species, and the decrease in RWC is the smallest, which greatly reduces the damage to the photosystem. After rehydration, there was no significant difference in Gs, but the Tr value was still significantly lower than that of the control group. With the exception of *Q. fabr,* there were no significant differences in the Gs values when comparing the four oak species with the control group. The results showed that the recovery ability of the *Q. serrata* photosystem was weaker than that of the other three oak species. *Q. serrata* may have weak repair ability after severe drought, which was indicated by the osmotic solutes recovery results.

### 3.5. Multivariate Statistical Analysis

The four oak species are distributed in different regions in China. In the long-term adaptation and evolution of the geographical environment, each species has a different level of drought resistance. The main differences are in the water parameters, osmotic regulation, antioxidant enzyme activity and photosynthesis. The changes in the physiological and biochemical indexes of the four oak species were complex and varied under drought stress, and they reach the peak in different time periods, which indicated that different oak species have different ways to adapt to drought stress. The PCA results showed that the boundary between the drought rehydration group and the control group was clear, and the PC1 components of the four oak species showed the same trend during the drought rehydration period, indicating similar adaptation to drought stress. PC1 mainly consisting of leaf water status, osmoregulatory solutes and photosynthesis, and correlation analysis shows that there is a significant correlation among them. At the same time, they are also closely related to antioxidant enzyme system and cooperate with each other to resist drought stress. Our results also showed that there were significant correlations between osmotic solutes and photosynthesis and between water parameters and antioxidant enzyme activities. The sensitivity of different oak species to drought stress is different, so the response time is also different, and the trees may be in different response stages simultaneously [55]. During continuous drought stress and rehydration, plants experience stress, adaptation, injury and repair. The comprehensive adjustment of different response mechanisms in different stress stages constitutes the overall drought resistance of plants [56]. The drought resistance of plants is the result of multiple factors. It is difficult to accurately and comprehensively reflect the drought resistance of plants with only one drought resistance index. The ranking results of a specific oak tree under different drought resistance indexes can vary greatly, so it is difficult to reasonably evaluate drought resistance according to a single index. Therefore, the weights of each indicator are calculated by PCA and then through the membership function method to comprehensively evaluate each index. The evaluation result can approximate the actual result.

## 4. Materials and Methods

### 4.1. Plant Material and Treatments

The seeds of *Q. fabri*, *Q. serrata*, *Q. acutissima* and *Q. variabilis* were collected from Wanzhou, Chongqing, China, in October 2020 (30°38′15′′ N, 108°37′21′′ E). No permission was required to collect these wild seeds. Then, the seeds were germinated in a sand bed, and the seedlings were transplanted to a light net bag container after rooting. The substrate of choice was a mixture of perlite and peat moss at a 1:3 ratio. The seedlings were cultivated in experimental greenhouses with automatic spray watering and natural light. Robust, disease-free seedlings were selected and transferred to a 21 cm × 16 cm × 21 cm (upper diameter × lower diameter × height) container in August 2021. One seedling was planted in each pot and filled with soil (5 kg). The cultivated substrate was nursery soil, the organic matter content was 54.07 g/kg, the fast nitrogen was 0.16 g/kg, the fast phosphorus was 0.04 g/kg, and the fast potassium was 0.16 g/kg. Routine field management of the planted seedlings was conducted without fertiliser. After one month of recovery, seedlings with similar growth were selected for natural soil drought stress and rehydration tests. Seasonal drought occurred frequently in Chongqing, China, and the choice of local oak species could better reflect the difference in response to seasonal drought stress.

Select seedlings with similar growth, one for each pot, 60 for each of the four oak species. They were randomly divided into a drought-rewatering treatment group (D-RW) and a well watered sample (CK), with 30 seedlings in each group. All pots were fully watered one day before the start of the treatment. The D-RW group was not watered again, lost water naturally and dried-out, while the CK group was watered normally. The D-RW group was rehydrated after one month. Leaf samples were taken at approximately 9:00 AM on days 1, 6, 11, 16, 21, 26, 31 and 36 of treatment, Ten plants were randomly selected in each group, and one leaf was taken from the same part. (3rd–5th from the tip of the branch down). The leaves were removed from the petioles, placed in self-sealing bags, numbered, and transported back to the laboratory in ice boxes at 0–4 °C. Five fresh samples were taken for water potential measurement, and the remaining parts were quickly ground with liquid nitrogen and stored in an ultralow temperature refrigerator at −80 °C for the determination of relevant physiological and biochemical parameters. Each indicator was repeated three times for each oak seedling species. Soil moisture was monitored using a TDR200 soil moisture sensor.

### 4.2. Leaf Relative Water Content and Water Potential

Five leaves were randomly selected for fresh weight (FW) measurement immediately after each sampling. Then, the leaves were placed in the dark in pure water for 24 h. The leaves were then removed to dry the water on the surface of the leaves and then were weighed, and the weight was recorded as the saturated fresh weight (TW). Finally, the leaves were placed in an oven at 85 °C and dried for 48 h to obtain a constant weight. Then, the leaves were removed and weighed, and the weight was recorded as the dry weight (DW) (Appendix A). The RWC of the leaves was calculated according to the formula: RWC(%) = ((FW − dW/TW − dW) × 100%) [57]. A WP4C dew point water potential metre (Psypro, Wescor Company, State of Utah, USA) was used to measure the leaf water potential. The water potential probe was clamped on each test sample, and rubber cement was added around the probe to form a closed space, which was the measurement chamber. The data were read every 5 min, and when the value was stable, the value was recorded 3 times to determine the average value, which was the leaf water potential.

### 4.3. Measurements of Osmotic Solutes

The Pro content was measured using the ninhydrin colorimetric method (G0111W Assay Kit, Suzhou Geruisi, China). The SS content was measured by the anthrone sulphate method (G0501W Assay Kit, Suzhou, China). The Soluble protein (SP) content was measured using the Komas Brilliant Blue G250 staining method (G0417W Assay Kit, Suzhou, China). The GB content was measured using the cycle colorimetric method (G0122W Assay Kit, Suzhou, China).

### 4.4. Measurements of Antioxidant Enzyme Activity and Malondialdehyde (MDA) Content

The SOD activity was determined by the nitrogen blue tetrazole (NBT) method (G0101W Assay Kit, Suzhou, China). The CAT activity was determined by sodium thiosulfate titration (G0105W Assay Kit, Suzhou, China). The POD activity was determined by the peroxidase chromatographic method (G0107W Assay Kit, Suzhou, China). The MDA content was determined by thiobarbituric acid colorimetry (G0109W Assay Kit, Suzhou, China).

### 4.5. Photosynthetic Parameters

After the experiment, the photosynthetic physiological parameters of the leaves were measured at 9:30–11:30 AM every five days. The upper, healthy, sunny-side leaves of the seedlings were observed using a LI-COR 6400 system (Li-COR, Lincoln, NE, USA). The air flow rate was set at 500 μmol/s, and the data were read after 2 min of stabilisation to record the net photosynthetic rate (Pn), transpiration rate (Tr) and stomatal conductance (Gs).

### 4.6. Data Analysis

All the data were tested for normality and Leneve’s test. The data were recorded and processed using Excel, and statistical analysis was carried out using SPSS 19.0. One-way analysis of variance (ANOVA) was used, and differences between treatments were analysed using Duncan’s multiple comparisons (*p* < 0.05). GraphPad Prism 9 and JASP 0.14.1 were used for plotting data. For PCA, the number of principal components is usually selected as the variable containing more than 80% information; that is, the cumulative contribution rate of characteristic roots should be greater than 80%. The drought tolerance of four oak species was evaluated by the membership function method. When the indicator was positively correlated with drought resistance, the formula was U(*X_ij_*) = (*X_ij_* − *X_j_*_min_)/(*X_j_*_max_ − *X_j_*_min_). When the indicator was negatively correlated with drought resistance, the formula was U(*X_ij_*) = 1 − (*X_ij_* − *X_j_*_min_)/(*X_j_*_max_ − *X_j_*_min_). The overall evaluation value was calculated as *X_i_* = ΣU(*p_m_* × *X_ij_*)/n. In the formula, *X_ij_* is the measured value of an index of a certain oak tree. *X_j_*_max_ and *X_j_*_min_ are the minimum and maximum values of the index. *p_m_* is the weight coefficient of the m-th principal component, and n is the number of indicators. These values were expressed as the mean ± standard error (SE) of three replicate samples.

## 5. Conclusions

This study simulated continuous drought stress that occurs under natural conditions. Changes in water parameters, osmotic regulation substance content, antioxidant enzyme activity and photosynthesis were observed dynamically. It was discovered that four oak species showed adaptive changes to drought stress and resisted early stages of drought stress by increasing the level of osmotic solutes and regulating the activity of antioxidant enzymes. However, with prolonged drought stress, all four oak species reached a tolerance limit. The water content, osmotic solutes content, antioxidant enzyme activity and photosynthetic parameters of the four oak species decreased to different degrees, especially at day 26. After rehydration, these indexes did not all recover to the control sample level, and the recovery ability of the four oak species was different. Multivariate statistical analysis showed that there were complex and close relationships between the leaf water status, osmotic solutes, antioxidant enzyme activities and photosynthesis, and they cooperated with each other to resist drought stress. PCA and the membership function method were used to analyse 13 physiological and biochemical indexes under continuous drought stress and rehydration. The drought resistance levels of the four oak species were as follows: *Q. serrata* > *Q. fabri* > *Q. variabilis* > *Q. acutissima*.

## Figures and Tables

**Figure 1 plants-11-00679-f001:**
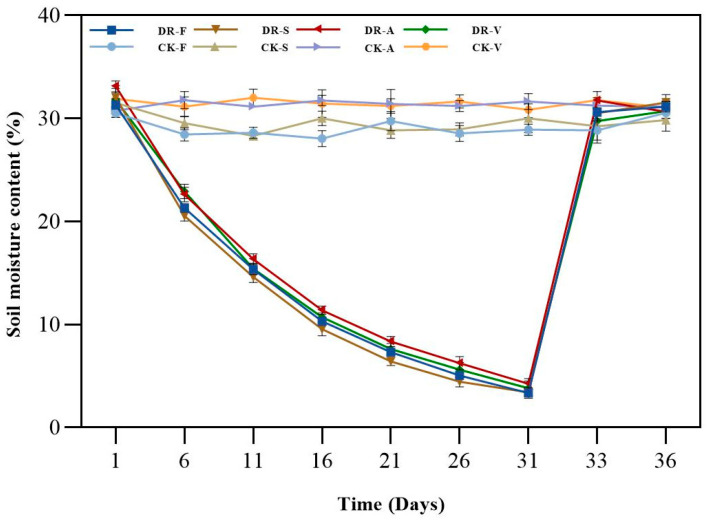
Effects of drought stress and rewatering on soil moisture content of four oak species (DR-F: drought-rewatering sample of *Q. fabri*; CK-F: well watered sample of *Q. fabri*; DR-S: drought-rewatering sample of *Q. serrata*; CK-S: well watered sample of *Q. serrata*; DR-A: drought-rewatering sample of *Q. acutissima*; CK-A: well watered sample of *Q. acutissima*; DR-V: drought-rewatering sample of *Q. variabilis*; CK-V: well watered sample of *Q. variabilis*).

**Figure 2 plants-11-00679-f002:**
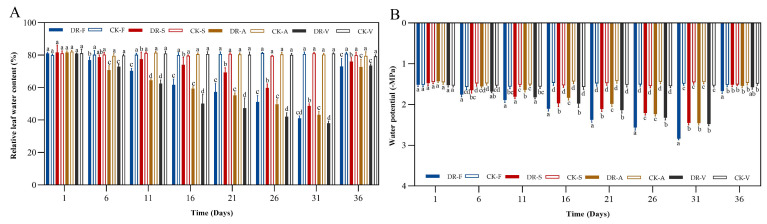
Effects of drought stress and rewatering on relative water content and water potential of four oak species. (**A**), relative water content (**B**), water potential. DR-F: drought-rewatering sample of *Q. fabri*; CK-F: well watered sample of *Q. fabri*; DR-S: drought-rewatering sample of *Q. serrata*; CK-S: well watered sample of *Q. serrata*; DR-A: drought-rewatering sample of *Q. acutissima*; CK-A: well watered sample of *Q. acutissima*; DR-V: drought-rewatering sample of *Q. variabilis*; CK-V: well watered sample of *Q. variabilis*. Different lowercase letters indicate significant (*p* < 0.05) differences among the four oak species subjected to the same treatment time.

**Figure 3 plants-11-00679-f003:**
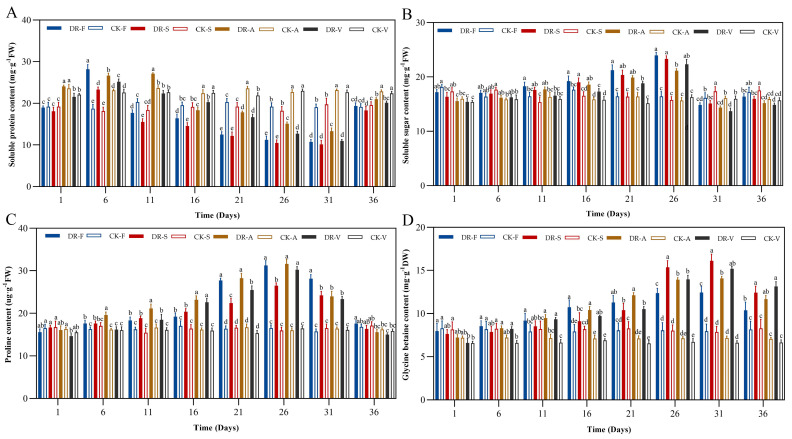
Effects of drought stress and rewatering on osmotic solutes content of four oak species (**A**) soluble protein, (**B**) soluble sugar, (**C**) proline, (**D**) glycline betaine. DR-F: drought-rewatering sample of *Q. fabri*; CK-F: well watered sample of *Q. fabri*; DR-S: drought-rewatering sample of *Q. serrata*; CK-S: well watered sample of *Q. serrata*; DR-A: drought-rewatering sample of *Q. acutissima*; CK-A: well watered sample of *Q. acutissima*; DR-V: drought-rewatering sample of *Q. variabilis*; CK-V: well watered sample of *Q. variabilis*. Different lowercase letters indicate significant (*p* < 0.05) differences among the four oak species subjected to the same treatment time.

**Figure 4 plants-11-00679-f004:**
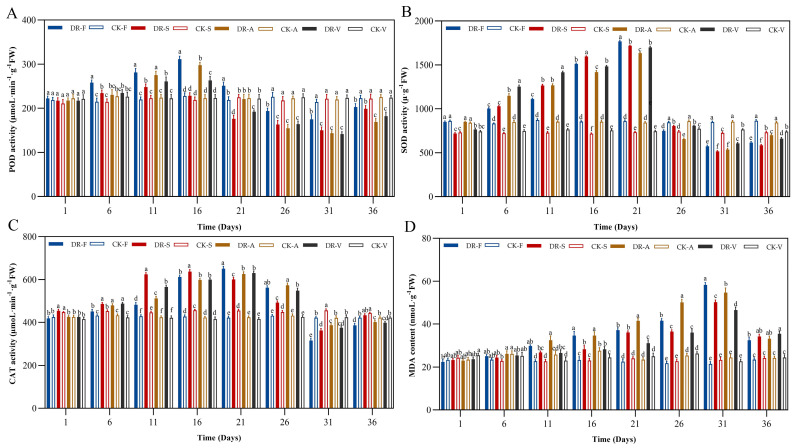
Effects of drought stress and rewatering on antioxidant enzyme activities and malondialdehyde of four oak species. (**A**) peroxidase activity, (**B**) superoxide dismutase activity, (**C**) catalase activity, (**D**) malondialdehyde content. DR-F: drought-rewatering sample of *Q. fabri*; CK-F: well watered sample of *Q. fabri*; DR-S: drought-rewatering sample of *Q. serrata*; CK-S: well watered sample of *Q. serrata*; DR-A: drought-rewatering sample of *Q. acutissima*; CK-A: well watered sample of *Q. acutissima*; DR-V: drought-rewatering sample of *Q. variabilis*; CK-V: well watered sample of *Q. variabilis*. Different lowercase letters indicate significant (*p* < 0.05) differences among the four oak species subjected to the same treatment time.

**Figure 5 plants-11-00679-f005:**
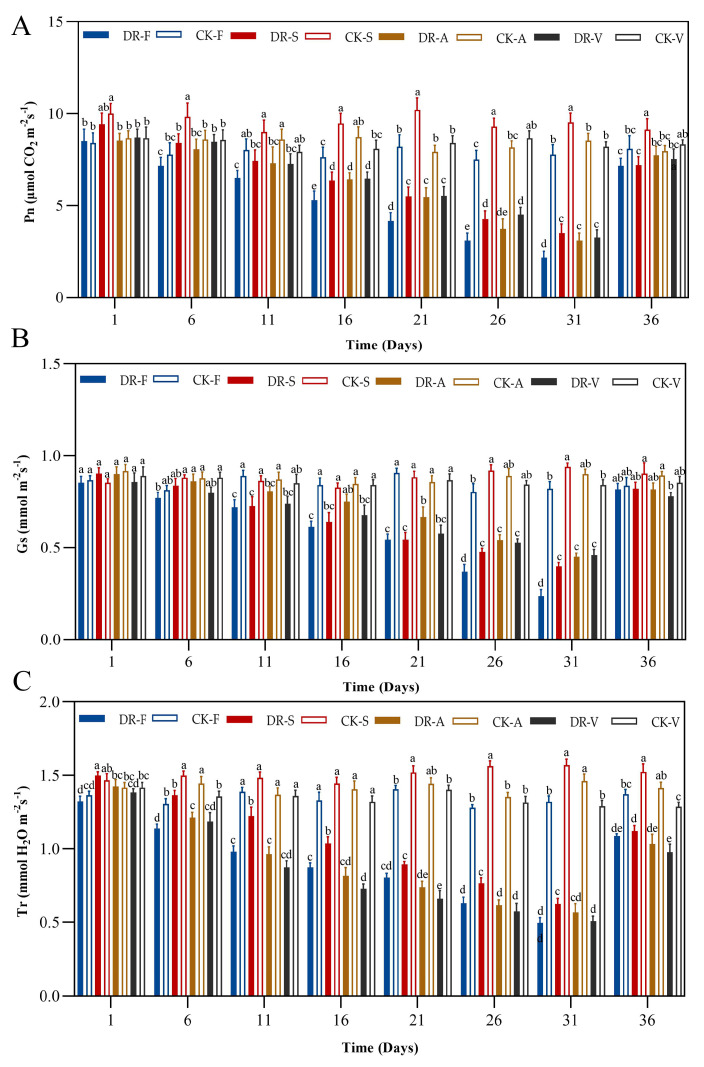
Effects of drought stress and rewatering on photosynthetic parameters of four oak species (**A**), net photosynthetic rate (**B**), transpiration rate (**C**), stomatal conductance. DR-F: drought-rewatering sample of *Q. fabri*; CK-F: well watered sample of *Q. fabri*; DR-S: drought-rewatering sample of *Q. serrata*; CK-S: well watered sample of *Q. serrata*; DR-A: drought-rewatering sample of *Q. acutissima*; CK-A: well watered sample of *Q. acutissima*; DR-V: drought-rewatering sample of *Q. variabilis*; CK-V: well watered sample of *Q. variabilis*. Different lowercase letters indicate significant (*p* < 0.05) differences among the four oak species subjected to the same treatment time.

**Figure 6 plants-11-00679-f006:**
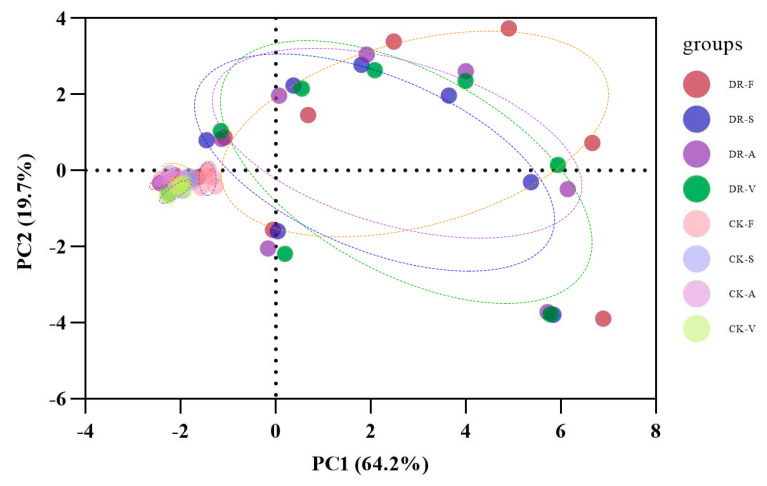
Principal component analysis of four oak species under continuous drought stress and rehydration conditions. DR-F: drought-rewatering sample of *Q. fabri*; CK-F: well watered sample of *Q. fabri*; DR-S: drought-rewatering sample of *Q. serrata*; CK-S: well watered sample of *Q. serrata*; DR-A: drought-rewatering sample of *Q. acutissima*; CK-A: well watered sample of *Q. acutissima*; DR-V: drought-rewatering sample of *Q. variabilis*; CK-V: well watered sample of *Q. variabilis*.

**Figure 7 plants-11-00679-f007:**
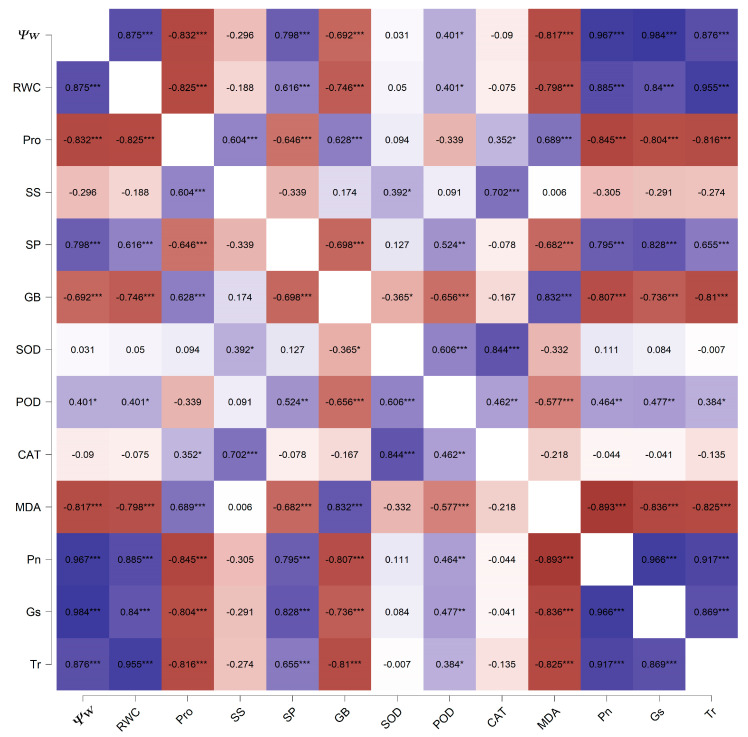
Correlation analysis among 13 physiological and biochemical indexes of four oak species under continuous drought stress and rehydration conditions. Ψw: Water potential; RWC: Relative water content; Pro: proline; SS: soluble sugar; SP: soluble protein; GB: glycline betaine; SOD: superoxide dismutase; POD: peroxidase; CAT: catalase; MDA: malondialdehyde; Pn: net photosynthetic rate; Tr: transpiration rate; Gs: stomatal conductance. * *p* < 0.05, ** *p* < 0.01, and *** *p* < 0.001.

**Table 1 plants-11-00679-t001:** Eigenvalue and cumulative contribution rate of each index of four oak species.

Measured Index	Principal Component	
	PC1	PC2
Relative water content	0.335	−0.014
Water potential	0.326	0.018
Soluble sugar	−0.174	0.365
Soluble protein	0.276	0.084
Glycine betaine	−0.309	−0.134
Pro	−0.322	0.099
SOD	−0.083	0.551
POD	0.150	0.447
CAT	−0.142	0.540
MDA	−0.311	−0.176
Pn	0.331	0.041
Gs	0.334	0.045
Tr	0.328	−0.024
Eigenvalue	8.352	2.562
Cumulative contribution rate (%)	64.24	83.94

**Table 2 plants-11-00679-t002:** Membership function values and evaluation index of the drought resistance of four oak species.

Item	*Q. fabri*	*Q. serrata*	*Q. acutissima*	*Q. variabilis*
Relative water content	0.3340	0.2399	0.2314	0.2325
Water potential	0.4529	0.5773	0.4193	0.3608
Soluble sugar	0.3002	0.2288	0.3272	0.2844
Soluble protein	0.0900	0.0883	0.0863	0.0823
Pro	0.1980	0.2321	0.3348	0.3919
SOD	0.0777	0.1053	0.0780	0.1094
POD	0.1384	0.0909	0.1013	0.0920
CAT	0.1125	0.1110	0.1236	0.1305
MDA	0.4770	0.5235	0.4702	0.5055
Pn	0.4322	0.4225	0.4932	0.5193
Gs	0.4354	0.4725	0.5329	0.4984
Tr	0.3608	0.3990	0.3285	0.3118
Comprehensive evaluation	0.2900	0.2983	0.2817	0.2819
Sequencing	2	1	4	3

## Data Availability

Not applicable.

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
