# Peer review of "Effects of Drought Stress and Rehydration on Physiological and Biochemical Properties of Four Oak Species in China"

_plants, 2022, doi:10.3390/plants11050679_

Round 1
Reviewer 1 Report
The manuscript “Effects of drought stress and rehydration on physiological and biochemical of four oak species in China” is based on the investigation of changes in physiological and biochemical properties of oak species under drought stress and then rehydration state. The research objectives were achieved by using systematic approaches for drought stress. It is valuable in the agriculture sector to understand the plant behavior under drought stress and then rehydration of the plants.
The language used in the manuscript is not up to the standard for publication, several major and minor mistakes are shown, and many of the phrases are not clear. It should go through a complete proofreading by a native speaker.
Title: The title is good and reflects the work actually done by the authors. However, it should add a word “properties” like physiological and biochemical properties of four oak species in China
Abstract: Abstract summarized the work precisely.
Keywords: Keywords are not appropriate; they should be different from title words.
Introduction
The introduction is technically fine and to the point, but there is a need to add information regarding the scope and requirements of the oak in the country. The introduction of the manuscript is quite lengthy, which is a good thing but on the other hand, it makes it look too exaggerated as well. The authors should make it concise and to the point because there are many irrelevant or outdated bibliographies in this section as well.
Materials and Methods
The methodology of the proposed study is fine and well-designed.
Results and Discussion
The other critical weakness of this manuscript is the Discussion part. It is not comprehensively described. In the discussion, some conclusions are too exaggerated. Authors should focus on specific and to the point information relating to their experiment and elaborate their achievements to justify their arguments developed in the results data. This section needs critical revisions
The conclusions section should also be a little more elaborative.
The references are not formatted as per the author’s guidelines provided by the journal.
To sum up, the manuscript can find interest among specialists when the comments will be taken into account.
Good Luck!
Reviewer 2 Report
The study by Xiong and co-Authors reports drought resistance traits of four different Quercus species. Authors recorded numerous data. However, the present version of the manuscript is quite confusing. As a result, the reading is not always fluent. Different sentences may be deleted because they are redundant and mask the most relevant recorded findings (in Specific comments are highlighted some sentences). The quality and the presentation of data must be improved. In the present form, comprehension of the data is very hard. As an example, you may improve Figs 2,3, 4 and 5 by choosing only 4 colours (one for each species) and identifying control samples by none pattern columns and drought-rewatered samples by filled pattern columns. This can permit to the reader to more promptly identify species and treatment. Figure 7 is illegible. Not last, a fall in estimation of the relative water content (RWC) exist. In response to dehydration, a loss of cell rehydration capability occurs (John et al., 2018, Plant Cell Environ; Trueba et al., 2021, Plant Cell Environ). As a consequence, especially under drought, RWC must be estimated as the ratio between the water content of the sample and the satured water contenta s recorded in well watered samples (Abate et al., 2021, Plant Physiol Bioch). Thus, it is expected that different RWC values were. The explanation of experimental planning must be improved. A supplementary figure might be helpful.
Some specific comments
L.36-37: Please, improve the sentence. The word “destroyed” may be changed witrh “severely affected” or “severely compromised”..
L.42: Please, improve the sentence.
L.51-52: The sentence is unclear. Please, improve the explanation about role of osmoregulation to face drought.
L.54: Please, delete the sentence “that is in close contact with the external environment”. Stem and roots are also in contact with the external environment!
L.62-64: There are a lot of similar experiments. Please, improve the references.
Figure 2: Please, check: MPa and not Mpa. Moreover, I suggest to change the scale of water potential in order to better highlight the decrease (i.e., from 0 -4 to 4 - 0)
L.150-172: Please, re-word this paragraph. Results are described in a very confusing way as well as order. Moreover, findings are not alwaysclearly described.
L.150-151: Please, check. I don’t see the described trend.
L.157-159: This sentence may be deleted.
- 161: respect to?
L.162-164: I suggest to avoid the word “normal” and to use “control samples”
L226: You may change the word “different” with “higher”
L.226-227, L.233, L.237: Please, delete these sentences. You have write these data above.
L.237: Please, check.
L.303-318: This paragraph must be reviewd on the basis of the right estimation of the relative water content. Moreover, Authors must take into account that the decrease in water status strongly depends on species-specfic stomatal control and cuticular tranpiration too.
- 420: Please, delete
L.436-444: The text must be re-wording (see comments above). Please, clarify: How much plants per species? How much leaves per plant and day sampling were removed? How much replicas per sampling? It is not clear. You performed different destructive measurements. Thus, relevant defoliation per plant may be occurred.
L.446; L.452: Please, specify better the collection and the estimation of samples for estimating RWC. Did you measure the fresh weight in situ? Moreover, as above underlined, the estimation must be re-calculate on the basis of recent literature data.
Round 2
Reviewer 2 Report
The study has been improved. However, some points need to be further reviewed. As a general comment, Results must be improved. Comments are rather twisted. Moreover, as underlined in the first revision, I strongly suggest to avoid the word “normal”. Authors may used “well-watered samples” or similar. This adjective is frequently and wrongly used in your paper. The word "normal" is misleading in an experimental irrigation planning.
Specific comments
Keywords: Including appropriate keywords in your paper helps indexers and search engines find your paper “Membership function” and “physiological parameter” are too much vague. I suggest to change them.
L.13: I suggest to change “normal watering” with “well watered samples (control group)”.
L.15: I suggest to change “osmotic adjustment substance content” with “osmotic solutes content”
L.38: You may change “affected” with “compromised”.
L.152: I suggest to change with: Osmotic solutes”
L.357: Please, see above
L.366: The sentence is not clear.Please, improve
L.473-474: Please, delete this sentence or choose to insert it here o in the Conclusions.
L.494: Please, improve the sentence
Figure 7: The quality of this figure is very low again.
Author Response
请参阅附件。
